# Effect of Onboard Training for Improvement of Navigation Skill under the Simulated Navigation Environment for Maritime Autonomous Surface Ship Operation Training

**Hyoseon Hwang** [1] 🆔, **Taemin Hwang** [1] 🆔 and **Ik-Hyun Youn** [2,*]

1 Department of Maritime Transportation System, Mokpo National Maritime University, Mokpo 58628, Korea
2 Division of Navigation & Information Systems, Mokpo National Maritime University, Mokpo 58628, Korea
* Correspondence: iyoun@mmu.ac.kr

**Abstract:** As the technology of the maritime autonomous surface ship (MASS) systems has geared toward autonomy, the importance of human operations in the shore control center (SCC) has gained in significance. Accordingly, the effects of the training method, including the traditional and new remote operator training methods have to be investigated in terms of MASS navigation safety. Therefore, this study conducted a comparative analysis to prove the effect of onboard training. The findings include the execution of a simulated navigation experiment, the extraction of rudder steering-related features, selection of significant features, and comparative analysis with network graph visualization. The separate results obtained from the "untrained" group and "trained" group were exhibited as the purpose of research for the effect of onboard training on navigation skills. Then, the authors interpreted the difference in each group allusively in accordance with features considering actual navigation and compared groups using descriptive statistics. Consequently, this study emphasized the importance of proving the effect of training before the new training technologies are used to train MASS remote operators in the future.

**Keywords:** maritime autonomous surface ship; remote operator; effect of training; simulated navigation experiments; comparative analysis

## 1. Introduction

With the technological development of the maritime autonomous surface ship (MASS) systems now gearing toward autonomy, it is expected that MASS operations will be carried out via interaction with the shore control center (SCC) under human operation via shore remote operators [1–5]. Before the commercialization of MASS, it is necessary to develop a new training method for remote operation [6–9]. Because human intervention cannot be entirely removed from the operation of autonomous ships, remote operators are identified to play a crucial role in the safe operation of MASS (as the autonomous navigation technology of MASS) [10,11]. As human factors account for the majority of maritime accidents, operators must be trained properly [12–15].

Navigation skills have been identified as the most important human factor in the training of traditional navigators [16]. At the moment, the training opportunities given to navigators are onboard training and SHS simulation training [17,18]. From onboard training, trainees can gain comprehensive ship operation knowledge and experience as an officer, such as ship management, cargo handling, risk management, navigation planning and execution, attitude, and leadership as an officer of the watch, including training for familiarity with the use of navigation equipment and a sense of life on board [15]. Although onboard training indeed includes both theoretical and practical ship handling training, these priceless opportunities are often difficult to provide regularly [19,20]. As a result of the research on maritime education and training, more simulation methods will be used for navigation training [21,22]. In future training, simulation training will account for a greater

proportion of navigator training than it does now, and the training must include navigation skill enhancement [23,24].

Existing navigation skills will have to be reinterpreted in line with the new education and training requirements for MASS [25,26]. In a similar manner, other research has emphasized the need for newly added training, such as the mandatory requirements of the future version of STCW [17,27]. Despite the growing debate over the new training, the expected effect of training on navigation skills is yet to be adequately elucidated. Thus, the verification of enhanced navigation skills ought to be conducted, especially before the new method of training is applied. As a result, this study conducted a simulation experiment on two different groups, namely, the "untrained group" and the "trained group," of onboard training as preparation for obtaining the newly added type of training. The proposed methods include related materials from the process of experiment execution to the interpretation of test results.

As a result, this study primarily aims to investigate the impact of onboard training on navigation skills improvement through a comparative analysis of various aspects of navigation evaluation perspectives.

## 2. Materials and Methods

The proposed methods use the data from the practical experiments. The most effective features in group separation were then applied to the network graph to determine the overall difference between the two groups using the navigation features extracted and selected based on the difference in onboard training. Figure 1 depicts the workflow of our proposed methods.

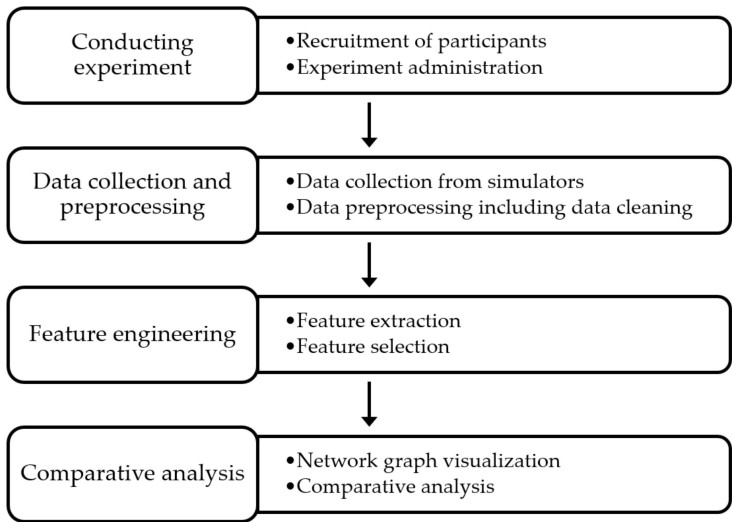

**Figure 1.** Workflow of the proposed methods.

This research conducted the simulation experiment designed and generated under contemplation. Then, the data collected from the experiment were preprocessed and analyzed for feature engineering. Afterward, selected features were scrutinized to compare data from the separated groups upon the onboard experience.

### 2.1. Design of Simulation Experiment

2.1.1. Participants

This research has recruited the students into two separate groups: "group A" and "group B". Group A consisted of students who had never experienced being on board the ship as cadets. Thus, their knowledge was limited to the theoretical education they acquired in class. On the contrary, group B consisted of students with onboard experience as cadets for one year on average. The specific information on participant recruitment is arranged in Table 1.

**Table 1.** Participants' information.

| Characteristics | Group A | Group B |
|---|---|---|
| Number of participants | 20 | 20 |
| Onboard training experience | None | One year |

### 2.1.2. Protocol

The authors created the basic navigation scenario solely to derive the different results of navigation skills, specifically the maneuvering skills of the groups, based on the simulation experiment design. Since groups differ significantly in terms of navigation experience, the experiment scenario used in this research was composed of only "route leg" and "waypoint" without territorial components. As a result, the participants' instructions were as simple as "navigate the ship on the route". The only permitted controller was the steering wheel. The experiment scenario is shown in Figure 2.

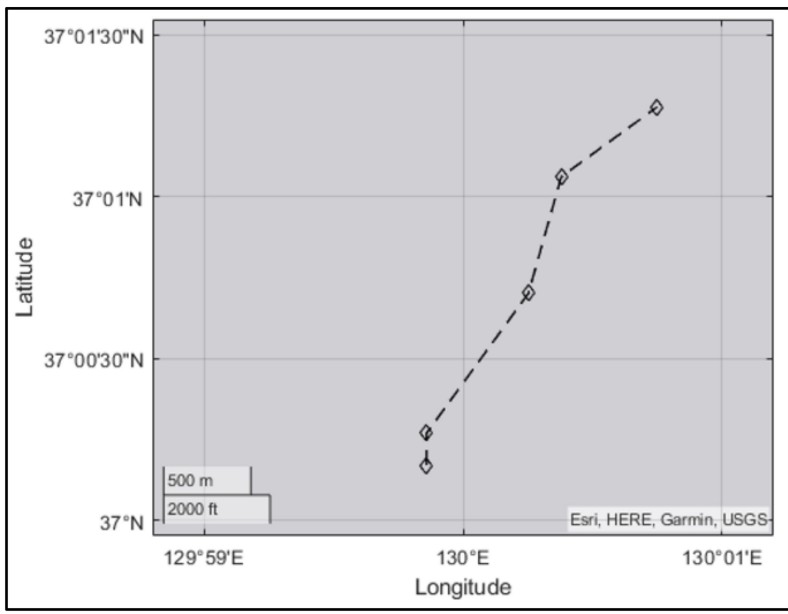

**Figure 2.** Experiment scenario.

The experiment used the full-mission-ship-handling simulator, and the target ship was the Mokpo National Maritime University training ship. The target ship is 133 m in length and 19.4 m in width and in 9196 GT; this ship is commonly boarded under the education curriculum.

### 2.2. Data Collection and Preprocessing

Data was collected from the simulator. The raw data consisted of 13 elements, as arranged in Table 2.

The raw data contained a temporal element, "Time" of elapsed time; the spatial element of "East" and "North" in the UTM datum; motion elements of "Heaving", "Yawing", "Pitching", and "Rolling" from 6 degrees of freedom; ship speed with RPM; and rudder degrees. Each value was represented numerically. Because the raw data was well-organized numerically, the data preprocessing focused primarily on unit conversion.

**Table 2.** Elements of raw data.

| Name | Sample | Unit |
|---|---|---|
| Time | $4 \times 10^{-1}$ | Second (s) |
| UTM (East) | $-6.8094 \times 10^{-4}$ | Meter (m) |
| UTM (North) | $1.1170 \times 10^{5}$ | Meter (m) |
| Heaving | $0.5142 \times 10^{-6}$ | Meter (m) |
| Yawing | $2.2744 \times 10^{-4}$ | Degree (°) |
| Pitching | $-1.1577 \times 10^{-7}$ | Degree (°) |
| Rolling | $4.1719 \times 10^{-4}$ | Degree (°) |
| Speed | 6.7110 | Meter per second (m/s) |
| Drift | $-1.0535 \times 10^{-3}$ | Meter per second (m/s) |
| Rate of turn | $2.0434 \times 10^{-3}$ | Degree per second (°/s) |
| Propeller (PORT) | $1.0287 \times 10^{2}$ | RPM |
| Propeller (STBD) | $1.0287 \times 10^{2}$ | RPM |
| Rudder angle | $-7.2000$ | Degree (°) |

*2.3. Feature Extraction*

Steering features were commonly related to maneuvering skills: the use of a rudder. The rudder is the primary device that allows the navigator to control the ship directly [28–30]. Thus, rudder movement-based features that can represent how the participant steered the ship are meaningful. Table 3 displays the extracted features.

**Table 3.** List of steering features.

| Name | Description | Unit |
|---|---|---|
| Max ROT | The maximum value of "rate of turn" | Degree per minute (°/min) |
| Mean ROT | The average value of "rate of turn" | Degree per minute (°/min) |
| STD ROT | Standard deviation value of "rate of turn" | Degree per minute (°/min) |
| Mean rudder | The average value of rudder angle | Degree (°) |
| First max rudder | Maximum rudder angle used for first altering | Degree (°) |
| Midship rudder | The temporal ratio of rudder angle at midship | Ratio (%) |
| Hard rudder | The temporal ratio of rudder angle over 30 degrees | Ratio (%) |
| Idle time | Idle time to use of rudder | Second (s) |
| First max time | Elapsed time to use "first max rudder" | Second (s) |
| Reverse time | Elapsed time to end first altering | Second (s) |

*2.4. Feature Selection*

Features selection was conducted based on the basic attribute features in the figures. A significant difference was identified from features in the shape of the box, whisker, and interquartile ranges based on aggregation and dispersion.

*2.5. Comparative Analysis*

Based on the findings, the comparative analysis focused on the difference in the onboard experience. This research used the network graph algorithm and conducted the interpretation to visualize the difference between the groups and explain the effect of onboard training on various aspects of navigation performance. The network graph algorithm is composed of "nodes", "edges", and "weight". When the weight of the edges is high, the connectivity between the nodes becomes strong. Furthermore, if the nodes are

strongly connected, the network is visualized as an aggregate shape as shown in Figure 3a, whereas the nodes are dispersed as a dispersion shape as shown in Figure 3b.

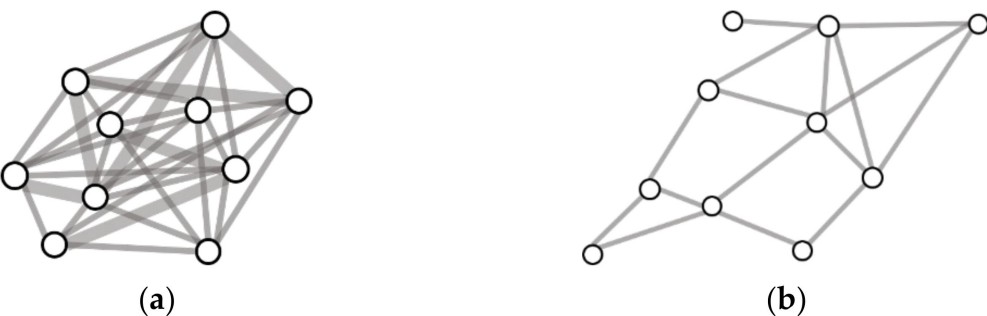

(**a**)       (**b**)

**Figure 3.** Concept of different shapes of networks: (**a**) aggregate shape and (**b**) dispersion shape.

This study defined each participant as a node, feature similarity as an edge, and absolute difference in feature value as a weight; this is known as connectivity. Then, edges with weak connections based on the average of weights were excluded from taking the verification of navigation skills into account.

## 3. Results

Steering features in Table 3 indicate fundamental skills, including the familiarization of rudder control. The authors then concentrated on examining the difference between features because rudder control is the most targeted attribute in the verification of enhanced navigation skills.

### 3.1. Result of Feature Extraction

The extracted feature demonstrated a significant difference as proof of onboarding training. Figure 4 depicts the differences between groups using normalized values of features, while Figures 5 and 6 depicts the aggregated navigation performance results of each group, which is useful for comparing the overall trajectories.

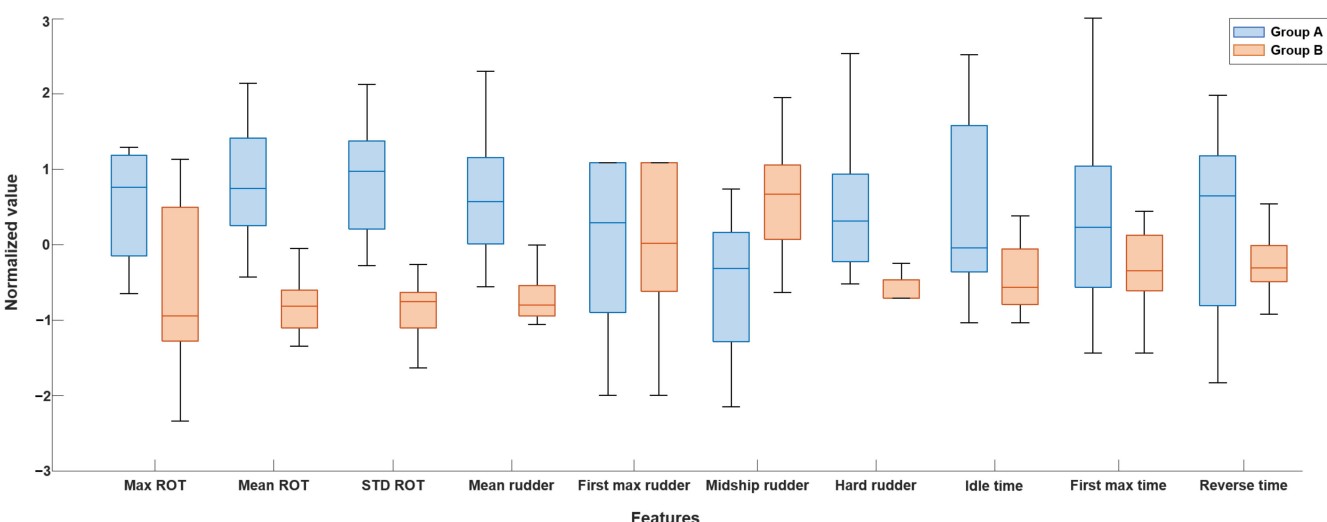

**Figure 4.** Maneuvering features and domains.

As proof of onboard training, the extracted feature was noted to have a remarkable difference. Figure 4 visualizes the difference of groups in each feature, and Figures 5 and 6 show the accumulated navigation performance results of each group, which is helpful in comparing overall trajectories.

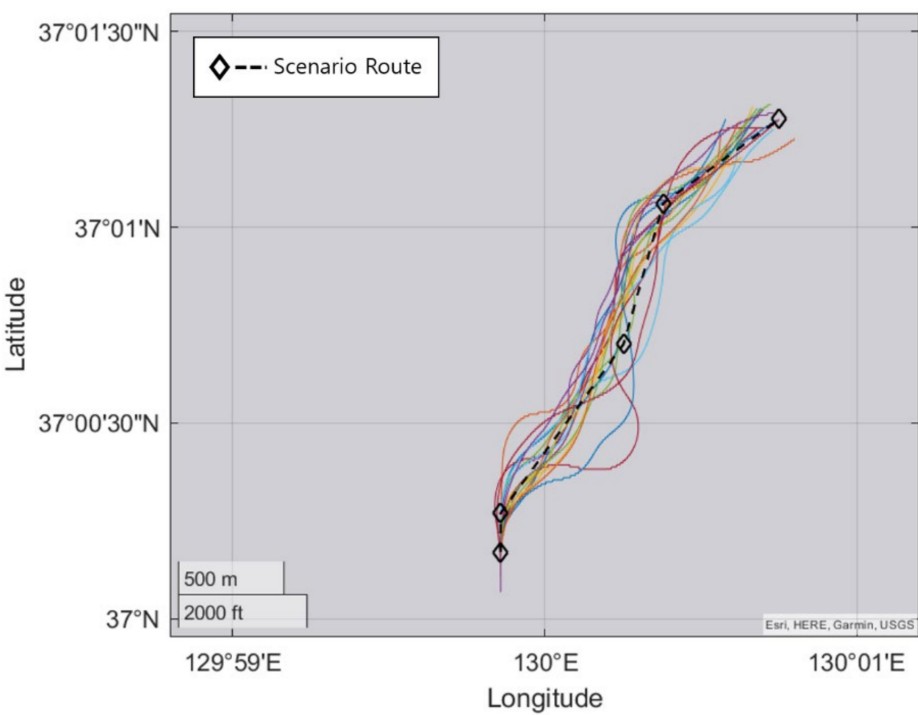

**Figure 5.** Navigation performance result: group A.

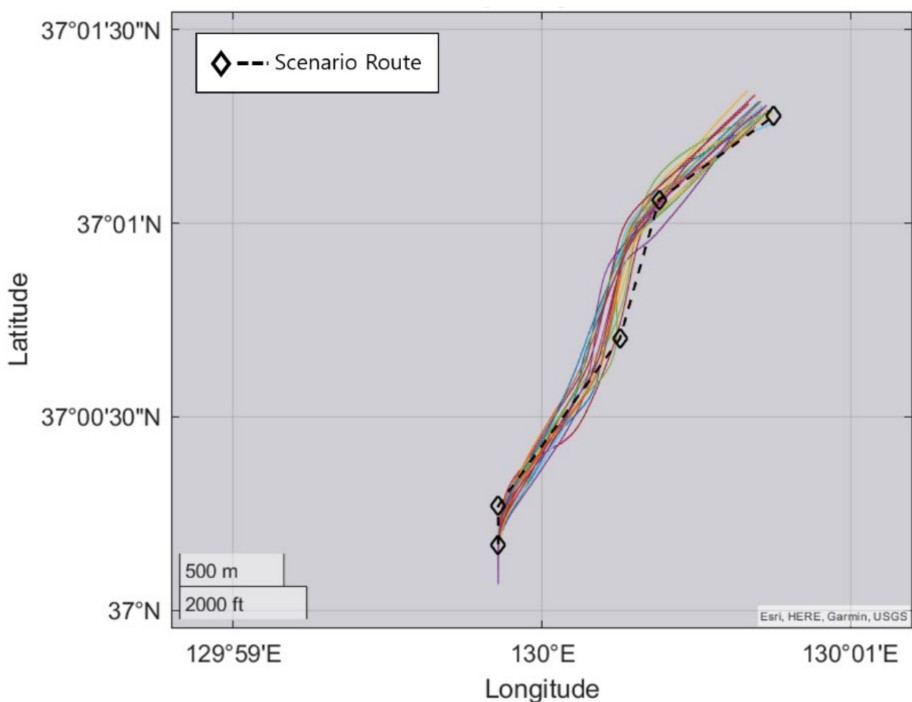

**Figure 6.** Navigation performance result: group B.

### 3.2. Result of Feature Selection

Figure 4 shows the basic boxplot attributes "box and whisker" and "interquartile range including skewness and dispersion. "First max rudder", "First max time", and "Reverse time" are the three features that have a non-significant difference in the shape of the box, whisker, and a form in which the interquartile ranges completely overlap in either direction. As the conformation displays, no particular discrepancy between the group was found. Another group of features that partially overlapped in box and whisker range were "Max

ROT", "Midship rudder", and "Idle time", yet their discrepancy was weak even when their skewness and dispersion differed. On the contrary, the "Mean ROT", "STD ROT", "Mean rudder", and "Hard rudder" showed a convincing difference between groups. Boxes were completely independent, and the interquartile ranges were disparate, with group A having large dispersion and group B having high skewness. As a result, features of "Mean ROT", "STD ROT", "Mean rudder", and "Hard rudder" were selected.

From the selected features, the authors found that "Mean ROT" and "STD ROT" are dependent; thus, the "STD ROT" was additionally excluded as Figure 7 shows final three selected features in the red frame.

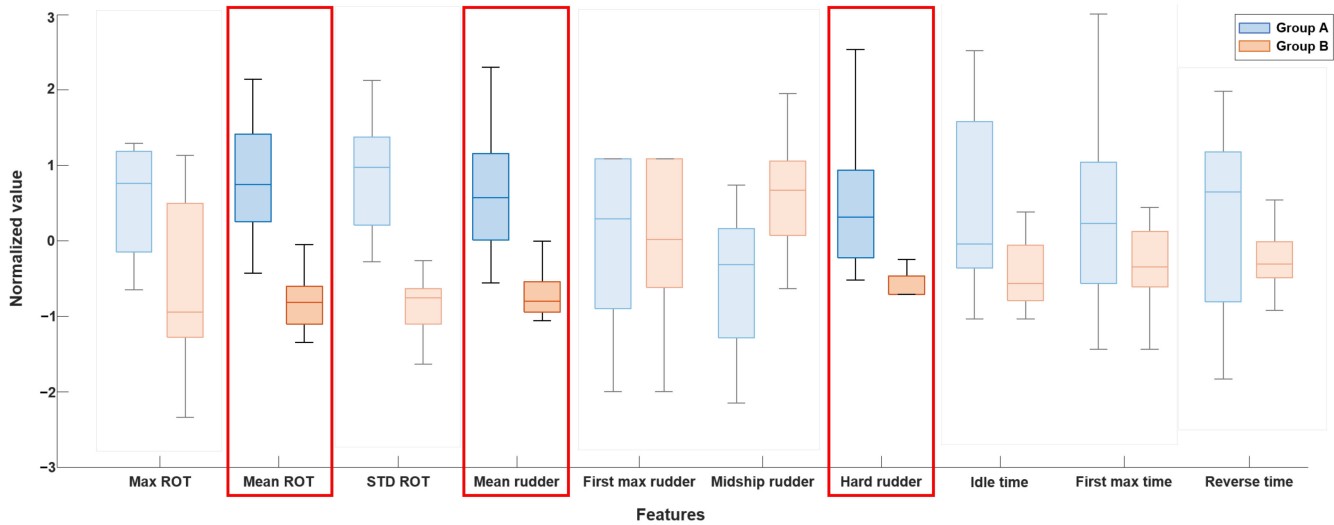

**Figure 7.** Selected features.

### 3.3. Comparative Analysis Result

Following the feature selection, this research calculated an absolute difference to represent the weight of edges. As a result, the closest connection had the lowest value, which is appropriate for visualization as connectivity. Then the maximum value was deducted from all values, and the values were transformed into absolute values again to reverse the meaning of the value, as a larger value indicates a stronger connection. As a result, in terms of weight, the reversed value came to mean what this research intended: larger values are stronger in connectivity, and smaller values are weaker. The network graphs of selected features are shown in Figures 8–10.

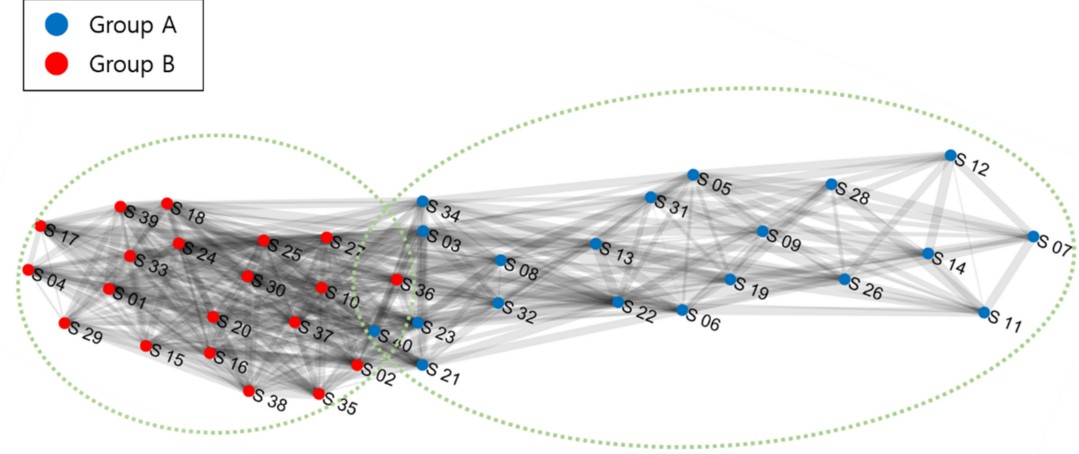

**Figure 8.** Network graph for the feature "Mean ROT".

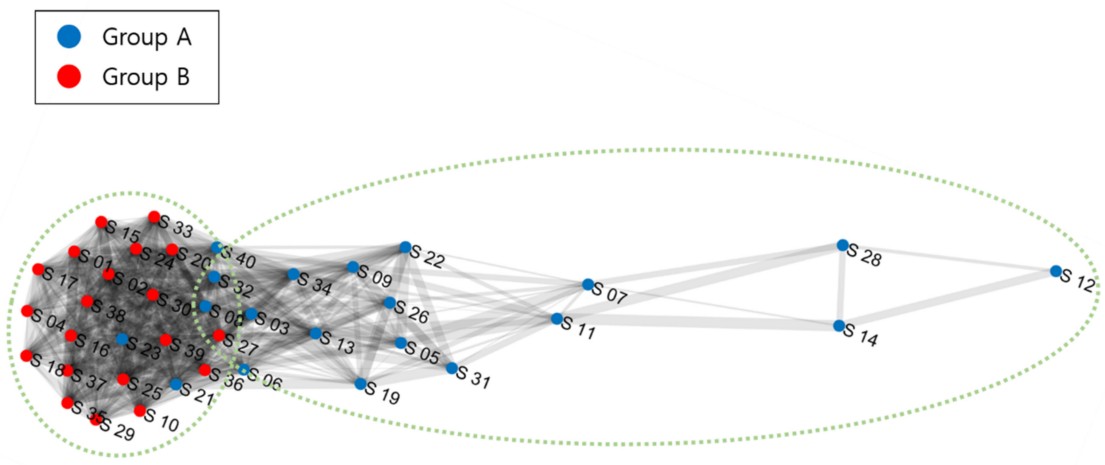

**Figure 9.** Network graph for the feature "Mean rudder".

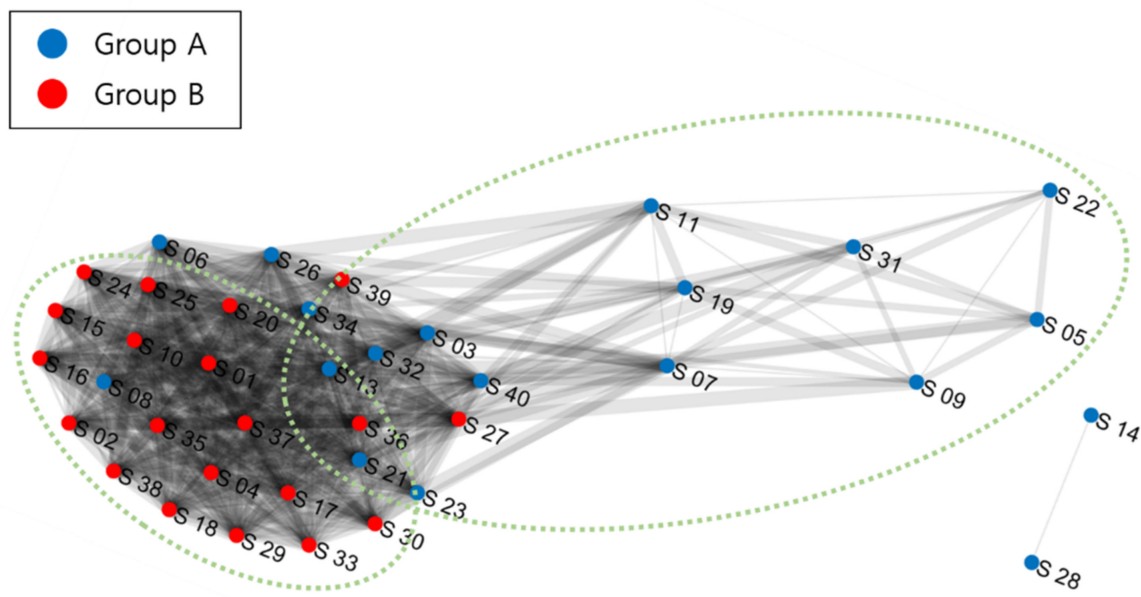

**Figure 10.** Network graph for the feature "Hard rudder".

## 4. Discussion

The authors chose features with notable differences based on the familiarization gained from onboard training. After some thought, it was discovered that the criteria for examining the features could vary along the set of factors in scenarios. As a result, when selecting features, we considered the designated factors of an experiment scenario. The network graph algorithm was then used in visualization to explain the significant differences. The obvious differences were thereafter discovered and explained.

### 4.1. Selected Features

The three preferred features are the most important aspect of rudder use. When the average rate of turn, also known as "Mean ROT", becomes excessive, the ship turns quickly, and maintaining course then becomes difficult. Similarly, the average use of rudder, that is, "Mean rudder", meant the participants swiped the rudder widely, reducing the ship's inertia of motion. Similarly, exaggerated control for the given scenario is a temporal ratio of rudder angle greater than 30 degrees. In terms of navigation skills, "Mean ROT" denotes an understanding of a ship's movements, "Mean rudder" denotes an understanding of the interaction between the rudder and the ship's movement, and "Hard rudder" denotes the

proclivity of participants in handling the ship. Those navigation skills are the fructification of the training, that is, the onboard training.

*4.2. Comparative Analysis of Groups*

4.2.1. Network Graph Interpretation

Although this research did not consider node phases, the examination focused on the shapes: "aggregation" and "dispersion" in network graph interpretation. The aggregation in Figure 3a depicts the network's strong bond. Those strong bonds were found in group B of Figures 8–10. Group B was described in the circle because its nodes were not precluded when edges in weak connection were removed. The network graph's strong connection indicates that the similarity was high, and the values of selected features were consistent within a certain range. Similarly, the dispersion shape was discovered in group A. The circle was crushed and scattered as a result of the preclusion process's severed edges, weakening the network's bond. In other words, there was a little resemblance.

4.2.2. Comparative Analysis

The difference of groups in the perspective of the network graph is well presented in Figures 8–10 with the interpretation in the previous section. Here, the authors specifically examined the selected features by using descriptive statistics in Table 4.

**Table 4.** Descriptive statistics for comparative analysis.

| Descriptive Statistics | Group A | Group B |
|---|---|---|
| Mean of "Mean ROT" | 29.5155 | 17.0955 |
| Median of "Mean ROT" | 29.1357 | 16.9824 |
| S.D of "Mean ROT" | 5.8821 | 2.8930 |
| Mean of "Mean rudder" | 13.0642 | 5.7998 |
| Median of "Mean rudder" | 12.3258 | 5.3612 |
| S.D of "Mean rudder" | 4.7802 | 1.4568 |
| Mean of "Hard rudder" | 0.1476 | 0.0143 |
| Median of "Hard rudder" | 0.1164 | 0 |
| S.D of "Hard rudder" | 0.1280 | 0.0252 |

In Table 4, the mean of "Mean ROT" depicts the entire trend of the specific feature, so the value shows how different the groups were in the comparison. The mean of group B was 0.58 times smaller than group A for the "Mean ROT", indicating that group B had a better understanding of a ship's movement. Similarly, knowledge of the interaction between the rudder and the ship's motion was far superior to group A, and participants' proclivity to handle the ship was reduced.

**5. Conclusions**

This research was motivated by the need to demonstrate the effectiveness of training, particularly the current training methods. When the era comes for remote operators to be trained and navigate the actual MASS, the training effect will be important for navigation safety. The study objectively identified the navigation skills that could be improved through onboard training in this study. A comparative analysis was conducted to demonstrate the value of onboard training. The participants in the experiment were divided into two groups based on whether or not they were trained onboard. The results of the analysis efficiently explain the difference between the groups in the specific features related to the use of the rudder. To emphasize the meaning of features in actual navigation, this study interpreted the difference in terms of the navigation skills. Although this research used a simple navigation scenario for the experiment and that participant recruitment was not systemic, the effect of onboard training was appropriately revealed. In the future, the training simulator designed explicitly for MASS remote control will be developed.

The authors expect that the remote operator training simulator will have better effects on improving operator navigation skills. Accordingly, the selected navigation performance features; "Mean ROT", "Mean rudder" and "Hard rudder" derived from this research are expected to be applied for the development of simulation performance-based navigation proficiency evaluation methods during the SRCO training period.

**Author Contributions:** Conceptualization, H.H., T.H. and I.-H.Y. Data curation, I.-H.Y. Formal analysis, H.H. and T.H. Funding acquisition, I.-H.Y. Methodology, H.H. and T.H. Project administration, I.-H.Y. Supervision, I.-H.Y. Visualization, H.H. and T.H. Writing—original draft, H.H. and T.H. Writing—review and editing, I.-H.Y. All authors have read and agreed to the published version of the manuscript.

**Funding:** This research and APC were funded by the Ministry of Oceans and Fisheries, Korea, through grant number 202106314.

**Institutional Review Board Statement:** The internal institutional review board determined to exempt the ethical review and approval for this study due to the ship handling simulator used in the experiment being a simulator used for education and training generally on the maritime institute. Also, the experiment was conducted in an environment without any physical restrictions on the subject's body.

**Informed Consent Statement:** Not applicable.

**Data Availability Statement:** Not applicable.

**Acknowledgments:** This research was a part of a project titled "Development of Smart Port-Autonomous Ships Linkage Technology," supported by the Ministry of Oceans and Fisheries, Korea.

**Conflicts of Interest:** The authors declare no conflict of interest.

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
