# Peer review of "Effect of Onboard Training for Improvement of Navigation Skill under the Simulated Navigation Environment for Maritime Autonomous Surface Ship Operation Training"

_applsci, doi:10.3390/app12189300_

Round 1

Reviewer 1 Report

Can you provide some details about the new training technology mentioned in the line 232 ?

Author Response

Dear. reviewer

Thank you for the great comments. The authors revised manuscript as per the comments, and arranged in attached file.

Thank you again.

Reviewer 2 Report

1. In order to establish the verification of enhanced navigation skills mentioned, I kindly recommend that previously included a brief point with your proposal for this skills. Them you can check with your results and incorporated all in the flow diagram, that I consider its necessary. Really I consider that it is very necessary to incorporate a new general flow diagram to clarify all technical paper.

2. Regarding Materials and Methods I have only comment, perhaps its necessary to increase the number of participants with another short group with more navigation expertise, because they have determined that only after 5 years the pilot acquires the minimum skills to operate alone.

3. The conclusions are a bit vague, they do not establish a clear criterion on what previous skills future candidates for remote EHS operators should have. 

I advise reinforcing this article on the points indicated, as well as emphasizing that it is a priority to establish the prior training and experience of the operators before their training, since the ability of these operators to respond quickly is an important and determining point in this process.

Also, I consider that it is an interesting article that is within the necessary quality parameters, so that once these changes have been incorporated it can be published.

Author Response

(The authors gave the same response as above.)

Reviewer 3 Report

This article presents a research study targeting the investigation of the impact of a proposed onboard training on the navigation skills improvement via a comparative analysis of varying navigation perspectives. A simple navigation scenario was performed on two groups of students with either without any experience or with some experience. The study appears to be technically sound, possesses merit. It is generally well-written. Authors are recommended to address the following comments:

- Specifics of the mentioned onboard training can be elaborated.

- In network graphs 8, 9, and 10, data labels for the nodes of importance may be explained / shown more clearly for better understanding of the graphs, especially at the intersections.

- Use of first person (i.e., "we", etc., used in Abstract and within text quite a few times) not recommended in this kind of technical research article. Third person and passive voice should be used.

- Line 73: students (plural)

- A proofreading is recommended.

Author Response

(The authors gave the same response as above.)
